# AI Assisted Medical Triage Assistance

## Abstract

In emergency and disaster scenarios, rapid triage is crucial to save lives. Standard protocols such as START and JUMPSTART provide structured guidance but are prone to human error under high cognitive load and time pressure. This paper presents an AI-assisted triage support system capable of analyzing video, audio, and textual inputs from virtual simulations to classify patients into urgency categories (red, yellow, green) in real time. We evaluate multiple models, including GPT-4o, GPT-4.1-nano, and o4-mini, and compare them on accuracy, inference speed, and confidence. Our results highlight the trade-offs in model selection and describe the potential of AI-human collaboration to improve training, reduce cognitive load, and increase decision making. Additionally, we examine scenarios where AI assistance is removed to assess user independence and over-reliance, providing insights for safe integration into real-world emergency response training.

## 1 Introduction

Medical triage is a base of emergency response, determining how patients are prioritized for treatment or disaster situations. Errors in triage can lead to severe consequences: under-triage may delay critical care for life-threatening conditions, while over-triage can waste limited resources and delay treatment for others. Protocols like START (Simple Triage and Rapid Treatment) and JUMPSTART provide structured approaches, but real-world decision-making is complex, and performance often suffers under stress and time constraints.

Recent developments in Artificial Intelligence (AI) and multimodal learning present new opportunities to enhance triage effectiveness. AI models can process and integrate diverse data flows, including images, audio, video, and textual patient information, to provide real-time decision support. By reducing the cognitive load and providing protocol-based guidance, AI can help standardize the triage outcomes and improve training efficiency.

This work introduces a comprehensive AI-assisted system designed for virtual triage simulations. Our contributions include: (1) development of a multimodal decision support framework capable of integrating video, audio, and textual data; (2) benchmarking multiple AI models to evaluate trade-offs between accuracy, speed, and confidence; and (3) preliminary exploration of human-AI collaboration, including the impact of AI withdrawal on user performance. Through this study, our objective is to train and explore AI works for emergency healthcare professionals.

## 2 Related Work

### 2.1 Rule-Based Triage Systems

START(1) and JUMPSTART(2) are widely adopted triage algorithms designed to categorize patients according to indicators such as breathing, circulation, and consciousness. While these rule-based systems are simple to apply and require minimal equipment, they are sensitive to user interpretation. Under high stress, variability in decision making is common, resulting in both under- and over-triage(3). These limitations highlight the need for tools that can assist practitioners in maintaining consistency and accuracy under pressure.

## 2.2 MACHINE LEARNING IN HEALTHCARE

Machine learning models have been increasingly applied to clinical settings, ranging from predicting patient deterioration to optimizing ICU resource allocation. Algorithms such as XGBoost(4) and random forests have demonstrated strong predictive performance for structured data, but their utility in real-time triage is limited without integration into multimodal data pipelines. Additionally, rigorous validation is required to prevent bias, false alarms, and unintended consequences in high-stakes healthcare scenarios.

## 2.3 AI IN SIMULATION AND TRAINING

Training systems use VR/AR for healthcare professionals to practice triage. Despite these advancements, many platforms lack automated feedback or adaptive decision support. (5). AI integration can improve realism, provide dynamic feedback based on user performance, and potentially increase skill acquisition. Key challenges are ensuring that AI outputs are interpretable and actionable for users.

## 2.4 LLMS FOR MEDICAL APPLICATIONS

Large language models (LLMs), such as GPT-4(6), have shown promise in clinical decision support, medical documentation, and knowledge retrieval. These models can provide protocol-based guidance, summarize patient information, and offer reasoning explanations. However, their deployment in healthcare settings demands careful oversight, robust validation, and safeguards to prevent errors, particularly in time-critical and high-risk situations.

## 3 METHODOLOGY

### 3.1 DATA COLLECTION

Our dataset is taken from virtual emergency simulation scenarios with multimodal inputs: video recordings capturing patient appearance and movement and text data describing patient condition or observed symptoms. Each scenario is labeled with a triage category: Red (immediate care required), Yellow (treatment can be delayed), and Green (minor injuries). Labels were validated by licensed physicians to ensure clinical accuracy.

### 3.2 DATA PREPROCESSING

The multimodal input requires alignment prior to training. So, we perform the below pre-processing steps:-

- **Image Preprocessing:** Video frames are resized to 224×224 pixels, with optional cropping, rotation, or brightness adjustments.
- **Audio Preprocessing:** Audio is resampled to a uniform sampling rate. This is done to ensure consistency across models
- **Textual Preprocessing:** Descriptions are tokenized and converted to vector embeddings to follow multimodal integration.

### 3.3 MODEL FRAMEWORK

The core model we used was GPT-4o, a multimodal transformer capable of balancing vision, audio, and language.This is capable of processing vision, audio, and language data. A Retrieval-Augmented Generation (RAG) pipeline supplements model predictions with protocol guidance from START and JUMPSTART documents. Three models were benchmarked:

- **o4-mini:** Lightweight, efficient, suitable for faster inference.
- **GPT-4.1-nano:** Optimized for ultra-fast processing.
- **GPT-4o:** High reasoning accuracy, slightly slower but more reliable.

The evaluation metric integrates multiple factors:

$$Score = f(A, \frac{1}{T}, C)$$

where $A$ = accuracy, $T$ = response time, and $C$ = confidence. Faster models are rewarded through $\frac{1}{T}$, while confidence indicates reliability of predictions.

### 3.4 BASELINE TESTING

Initial testing used sample simulation data to establish baseline performance, providing a reference for further optimization and feature augmentation.

### 3.5 GROUND-TRUTH DATA

All triage labels were verified by physicians within the context of the 60 Seconds to Survival simulation game. This ensures that evaluation metrics are grounded in clinically valid assessments.

## 4 EVALUATION

### 4.1 DATASETS

The evaluation dataset consists of multiple virtual triage scenarios, including video, audio, and text inputs. Each scenario simulates time-critical conditions with multiple patients, reflecting realistic decision-making pressures and environmental variability.

### 4.2 EVALUATION METRICS

We assess models across several metrics:

- **Accuracy:** The number of accurate predictions assigned to correct triage predictions.
- **Response Time:** Measured from the input flow to the output of the model.
- **Confidence Score:** A value that describes how strongly the model supports its own predictions.

Weighted error:

$$E = \beta \cdot U + (1 - \beta) \cdot O$$

where $U$ = under-triage rate, $O$ = over-triage rate, and $\beta > 0.5$ prioritizes patient safety.

### 4.3 HUMAN-AI COLLABORATION

We analyze three experimental conditions:

1. **Without AI:** Users rely on training alone. Provides a baseline for independent skill and cognitive load. Pros: It leads to independent decision-making and baseline skills. Cons: It will increase cognitive load, slower decision-making, and risk of errors in complex scenarios.

2. **With AI:** Real-time guidance supports decision-making, reduces cognitive load, and can provide reasoning explanations. Pros: This leads to increase in accuracy, reduces cognitive load, and provides reasoning explanations and feedback. Cons: Could lead to risk of over-reliance on AI and possible complacency in developing independent reasoning skills.

3. **AI withdrawn mid-task:** Evaluates how dependent users become on AI, revealing strengths and potential over-reliance. Pros: This process can reveal the user's understanding of rules and protocols. Cons: This can lead to short-term reduction in task performance.

Although we have not yet conducted full experiments, preliminary simulations and pilot observations suggest that AI-assisted decisions could improve accuracy and reduce cognitive load during

triage training. We also anticipate that removing AI assistance may reveal potential over-reliance, highlighting the importance of safeguards in AI integration. Additionally, initial observations indicate that explanations retrieved via RAG could enhance trainees' understanding of the reasoning behind triage protocols. Ongoing work aims to systematically quantify these effects to better inform the design of AI-supported training environments.

### 4.4 CURRENT STAGE

Currently, we are working with simulated dataset to assess feasibility, benchmark models, and understand the trade-offs between speed, accuracy, and confidence. These results provides baseline evaluation and guidance for future expansion to real-world data and model fine-tuning.

## 5 FIGURES

The following three figures explain the core components of our project:-

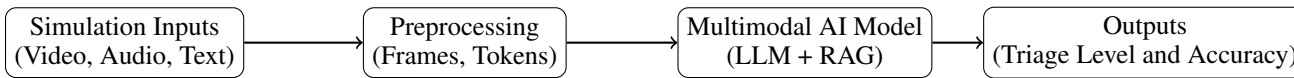

Figure 1: System architecture of the AI-driven triage decision support system.

Figure 1 illustrates the overall workflow of our AI-driven triage system. Simulation inputs such as video, audio, and text, are first pre-processed into standardized formats, such as extracted video frames and tokenized text. These features are then passed to a multimodal model that integrates large language models (LLMs) with retrieval-augmented generation (RAG). The system produces triage predictions (red, yellow, green) which explains the urgency levels.

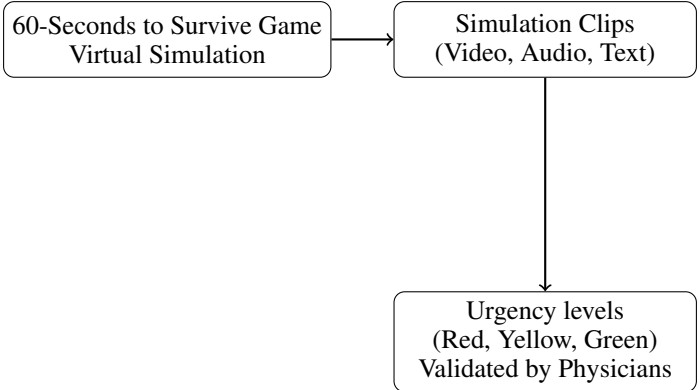

Figure 2: Dataset from survival game with physician-validated labels.

This figure no.2 illustrates how the data set is built from the '60 Second Survival' simulation game. Each round of games generates multimodal clips that contain video, audio, and textual information on patient scenarios. The clips are annotated with ground-truth triage labels (red, yellow, green), which were validated by licensed physicians. These annotations provide the benchmark for evaluating the model performance.

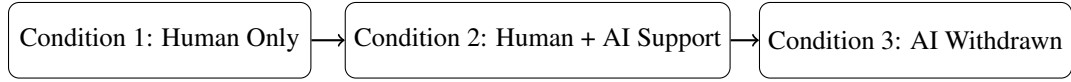

Figure 3: Study design for human-AI collaboration.

Figure no.3 outlines the experimental design of human-AI collaboration in triage. We explain three conditions which are: (1)Without AI, only human decision, (2) human support with AI suggestions,

and (3) AI withdrawn during mid-task. This evaluation setup enables us to check how AI affects trust, accuracy, and adaptability and flexibility under stress.

## 6 TABLES

Table 1: Model performance on virtual triage simulations

| Model | Accuracy (A) | Response Time (T, s) | Confidence (C) |
|-------|--------------|----------------------|----------------|
| o4-mini | 0.667 | 3.9 | 0.752 |
| gpt-4.1-nano | 0.642 | 0.94 | 0.721 |
| gpt-4o | 0.655 | 2.8 | 0.836 |

Based on the above table no.1 , we can observe the performance summary:

- **Accuracy:** o4-mini achieves the highest, showcasing efficient multimodal integration. GPT-4.1-nano trades some accuracy for speed, while GPT-4o balances reliability and processing time.
- **Response Time:** GPT-4.1-nano is fastest; o4-mini is slower but still practical; GPT-4o is intermediate.
- **Confidence:** GPT-4o exhibits highest prediction confidence, supporting its use in high-stakes scenarios.

These findings from table no.1 highlight that the choice of model should depend on the operational choices, whether speed or safety is the higher priority. In future work, we plan to validate the system on real-world data, extend the simulation environment, and explore ensemble approaches.

## 7 CONCLUSION

This work presents a multimodal AI system for medical triage that integrates video, audio, and text inputs. Different AI models demonstrate trade-offs between accuracy, speed, and confidence, highlighting the importance of careful model selection. Preliminary human-AI collaboration results suggest AI support can enhance training effectiveness and decision reliability while exposing potential over-reliance risks.

## 8 FUTURE WORK

The following steps aim to improve disaster readiness in triage training and potentially in real-world disasters

- Validating models on real-world patient data.
- Designing metrics that prioritize under-triage over over-triage.
- Developing edge-deployable AI models for offline and low-resource settings.

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

## 9 ACKNOWLEDGMENTS

We acknowledge the use of AI tools for minor writing assistance in this paper. These tools were used only to rephrase sentences and improve language clarity.

