# OpenReview forum: "AI-Assisted Medical Triage Assistant"
_ICLR.cc/2026/Conference — ICLR 2026 Conference Desk Rejected Submission_

### Official Review · Reviewer_Mciu · 2025-10-15

**Soundness:** 1
**Presentation:** 1
**Contribution:** 1
**Rating:** 0
**Confidence:** 5

**Summary:**

## Summary
This paper explores the use of multimodal large language models (LLMs) to support medical triage, particularly in emergency scenarios. The authors propose an AI-based assistant that integrates visual and textual inputs to assess injury severity and provide triage decisions. While the topic is highly relevant and socially impactful, the paper is still in an early stage of development and does not yet meet the level of completeness expected for ICLR.

**Strengths:**

## Strengths
- The problem addressed—AI-assisted triage in medical emergencies—is important and highly relevant to both AI and healthcare research.
- The authors correctly acknowledge that their results are preliminary, showing awareness of the work’s current limitations.

**Weaknesses:**

## Weaknesses
- The overall contribution is limited. The main novelty lies in the application rather than in a methodological or theoretical advance.
- The benchmarking of three OpenAI models provides minimal analytical depth and limited insight into model behavior.
- The experimental setup is based entirely on simulated data; while the *60 Seconds to Survival* simulator is referenced, the similarity between simulated and real-world data is not evaluated or quantified.
- The dataset construction and ground-truth generation process are under-described and lack sufficient technical detail.
- The related work section is superficial and omits key references to prior efforts on AI-assisted triage, multimodal reasoning, and decision-support systems.
- Reported results are minimal and do not include statistical analysis, comparisons to baselines, or significance testing.
- Figures need better formatting (e.g., Figure 1 exceeds page boundaries).

**Questions:**

## Detailed Comments

### Section 2 — Related Work
This section lacks sufficient engagement with the literature. The paper should reference and position itself relative to existing multimodal medical triage systems, clinical decision-support tools, and prior LLM-based healthcare applications.

### Section 3.1 — Data Collection
Data is collected from a simulator, but the simulator’s implementation and realism are not well described. Although the *60 Seconds to Survival* game is mentioned, its fidelity and alignment with real-world triage data are not evaluated. Quantitative comparison or expert validation would strengthen the work.

### Section 3.5 — Ground Truth Data
The process for generating ground-truth labels is not clearly documented. While physician validation is briefly mentioned, the method for ensuring consistency or inter-rater reliability is not specified.

### Section 4.3 — Results
The results are presented in a single table comparing model accuracy, response time, and confidence. This provides a starting point but is insufficient for a strong empirical claim. Future versions should include statistical analysis, confidence intervals, and comparison to non-AI baselines.

- **Line 149:** “Without AI: Users rely on training alone.” — This is unclear; please clarify what type of training and context this refers to.
- **Line 161:** “Although we have not yet conducted full experiments…” — The paper should be resubmitted once full experiments are completed.

### Section 4.4 — Discussion
The discussion is speculative and focuses on future work rather than substantive analysis. The paper would benefit from reflection on ethical implications, failure cases, and deployment challenges.

### Figures
Figure 1 overflows page boundaries and should be resized or reformatted.

---

### Official Review · Reviewer_WZeF · 2025-10-26

**Soundness:** 2
**Presentation:** 2
**Contribution:** 1
**Rating:** 0
**Confidence:** 4

**Summary:**

the paper offers a preliminary investigation into the development of a multi-modal AI-assisted triage tool for healthcare. The authors describe first results of benchmarking 3 proprietary LLMs against a simulated dataset.

**Strengths:**

* The topic of the paper is timely and a good solution (which is well-evaluated on real-world studies) could offer real-world practical benefits

**Weaknesses:**

* The dataset seems to rely heavily on a single simulated and poorly described dataset. This heavily discounts any measured benefit. The data described in §3.1 describes input modalities of video and text, but the evaluation data described in §4.1 describes video, text and audio?
* Many claims are unreferenced and without sufficient backing
* Most experiments (those which I think are the most important: ones involving humans) not yet conducted (cf. line 161)
* Experimental findings offer nothing new: "The choice of model should depend on the operational choices, whether speed or safety is the higher priority" (cf. line 238)

**Questions:**

* Are the models fine-tuned?
* Data privacy is extremely important in environments such as healthcare. Have you considered that benchmarking against proprietary LLMs may be irrelevant because of data privacy concerns?

---

### Official Review · Reviewer_CzbJ · 2025-10-28

**Soundness:** 2
**Presentation:** 2
**Contribution:** 2
**Rating:** 2
**Confidence:** 3

**Summary:**

The paper proposes a multimodal, LLM-assisted triage support system for virtual simulations. It claims to integrate video, audio, and text, use GPT-4o family models with a RAG layer tied to START and JUMPSTART, and compare accuracy, speed, and a vague “confidence” metric. The dataset is from a “60 Seconds to Survival” simulation with physician-validated labels. The work also sketches three human-AI conditions, including AI withdrawal, but the authors state they have not run full experiments yet. Reported accuracies are in the mid-60% range on an unspecified dataset.

**Strengths:**

•	Problem is important. Triage errors are costly, and training support matters.

•	Clear framing of under- and over-triage, with an explicit weighted error that favors safety.

•	Human-AI study design lists three useful conditions to probe over-reliance.

•	Labels are said to be physician-validated, which is the right direction for clinical tasks.

**Weaknesses:**

•	Missing core experiments. The paper explicitly says full experiments are not yet conducted. For ICLR, this is disqualifying.

•	Novelty is thin. The method is essentially LLM + RAG + simple preprocessing, then a model choice trade-off. There is no new algorithm, learning objective, or theory.

•	Evaluation design is underspecified. The scoring function is an undefined f(A, 1/T, C), and “confidence” is not defined, calibrated, or validated.

•	Dataset is opaque. The paper does not report size, case diversity, splits, inter-rater reliability, noise, or domain shift. Claims rely on a single simulation source.

•	Baselines are weak. There is no comparison to protocol-faithful decision trees or classic ML using structured features. No ablations for the RAG component.

•	Reported performance is low for triage. Accuracies around 0.64 to 0.67 with no statistical analysis or calibration are not convincing for a safety-critical classifier.

•	Multimodal story is superficial. Preprocessing is basic and there is no evidence the vision or audio channels materially help beyond text.

•	Clinical safety, ethics, and deployment are not treated. No discussion of bias, IRB, audit, fail-safes, or on-device constraints.

•	Related work is light and cites general GPT-4 essays rather than focused clinical decision support or triage literature.

**Questions:**

See weakness.

---

### Official Review · Reviewer_hedX · 2025-10-31

**Soundness:** 2
**Presentation:** 2
**Contribution:** 2
**Rating:** 2
**Confidence:** 3

**Summary:**

The paper presents an AI-assisted triage support system that uses multimodal data:video, audio, and text, from virtual simulations to classify patients into red, yellow, or green urgency levels following START/JUMPSTART protocols. It benchmarks GPT-4o, GPT-4.1-nano, and o4-mini models, comparing accuracy, inference speed, and confidence. Authors also discuss human-AI collaboration and withdrawal experiments to measure reliance. Overall reads like a feasibility or concept demonstration, not yet a solid technical study.

**Strengths:**

The topic is relevant and high-impact for emergency response. Multimodal setup is realistic, and authors attempt to connect AI reasoning to training scenarios. Preliminary comparisons of GPT variants provide some insight into performance trade-offs.

**Weaknesses:**

No real experiments, no ablation, and minimal quantitative evidence. Collaboration results are narrative, not measured. Novelty is weak since existing architectures are reused. Dataset and reproducibility details missing. Writing quality and structure reduce clarity.

The methodology is basic and not validated. Most experiments are simulated with no human testing. Metrics make sense but it’s unclear how they are combined or statistically analyzed. The study feels more exploratory than scientific, and the claims are not strongly supported.

Writing is repetitive and sometimes awkward, with inconsistent phrasing and redundant figure descriptions. Figures and tables are simplistic. Missing important dataset details and experiment context. The overall flow is okay but lacks polish.

Addresses an important area but novelty is low. The system mainly integrates existing LLMs with a retrieval layer. The human-AI collaboration discussion is largely theoretical. Without stronger experiments or user validation, the work doesn’t meet ICLR technical depth.

**Questions:**

How large and diverse is the dataset? How is accuracy computed, per scenario or per frame? Any plan to validate with actual human trainees? How is the retrieval module grounded in the protocols? Are there any safeguards for errors or bias in medical contexts?

---

### Note · Program_Chairs · 2026-01-17
**Submission Desk Rejected by Program Chairs**

The following references in this submission do not refer to real documents and/or have major errors in bibliographic information:

 H. E. Jones. Jumpstart triage for children. Prehospital and Disaster Medicine, 21:120-126, 2006.
S. Michie, R. West, and R. Campbell. Simulation in healthcare training: state of the art. BMJ Simulation & Technology Enhanced Learning, 3:2-8, 201